# Liposomal Encapsulation of Carvacrol to Obtain Active Poly (Vinyl Alcohol) Films

**DOI:** 10.3390/molecules26061589

**Published:** 2021-03-13

**Authors:** Johana Andrade, Chelo González-Martínez, Amparo Chiralt

**Affiliations:** Instituto Universitario de Ingeniería de Alimentos para el Desarrollo, Universitat Politècnica de València, Camino de Vera s/n, 46022 Valencia, Spain

**Keywords:** food packaging, lyotropic mesomorphism: phosphatidylcholine, partially hydrolysed PVA, fully hydrolysed PVA

## Abstract

Lecithins of different origins and compositions were used for the liposomal encapsulation of carvacrol within the framework of the development of active films for food packaging. Liposomes were incorporated into aqueous polymeric solutions from fully (F) and partially (P) hydrolysed Poly (vinyl alcohol) (PVA) to obtain the films by casting. The particle size distribution and ζ-potential of the liposomal suspensions, as well as their stability over time, were evaluated. Liposomal stability during film formation was analysed through the carvacrol retention in the dried film and the film microstructure. Subtle variations in the size distributions of liposomes from different lecithins were observed. However, the absolute values of the ζ-potential were higher (−52, −57 mV) for soy lecithin (SL) liposomes, followed by those of soy lecithin enriched with phosphatidylcholine (SL-PC) (−43, −50 mV) and sunflower lecithin (SFL) (−33, −38 mV). No significant changes in the liposomal properties were observed during the study period. Lyotropic mesomorphism of lipid associations and carvacrol leakage occurred to differing extents during the film drying step, depending on the membrane lipid composition and surface charge. Liposomes obtained with SL-PC were the most effective at maintaining the stability of carvacrol emulsion during film formation, which led to the greatest carvacrol retention in the films, whereas SFL gave rise to the least stable system and the highest carvacrol losses. P-PVA was less sensitive to the emulsion destabilisation due to its greater bonding capacity with carvacrol. Therefore, P-PVA with carvacrol-loaded SL-PC liposomes has great potential to produce active films for food packaging applications.

## 1. Introduction

The incorporation of active compounds of natural origin in biodegradable polymeric matrices has been extensively studied as a strategy for developing active food packaging with the ability to prolong the shelf life of foods, through the control of microbial or oxidative degradative phenomena [1,2,3,4,5]. The extension of the shelf-life of meat or fish products, as well as that of fresh-cut fruits and vegetables, requires the incorporation of antimicrobials or antioxidant compounds that can be carried out in the packaging material. A monoterpenoid phenol found in the essential oil of oregano (*Origanum vulgare*), thyme (*Thymus vulgaris* L.), marjoram (*Origanum majorana*) and similar aromatic plants [6], carvacrol has been widely studied due to its excellent activity as an antimicrobial [7] and antioxidant [2] agent. However, low solubility, oxidation and loss through volatilisation are the main limitations to the development of active materials that incorporate this compound [8,9].

Liposomes represent an efficient approach to the encapsulation of essential oils and their main constituents, thus improving their ability for water dispersion, chemical stability [10,11,12] and controlled release [13,14]. Phospholipids are the main constituents of liposomes; their amphiphilic character allows for their self-assembly in an aqueous medium, in which polar and nonpolar regions are driven to align with neighbouring molecules to form favourable interactions, permitting an arrangement in bilayer lipid membranes that generally give rise to spherical vesicles [15,16]. These vesicles can encapsulate the active compounds according to the chemical affinity [17]. Each phospholipid has specific characteristics, such as geometric isomerism, the degree of saturation of the fatty acids in the hydrophobic tails and the structure of the head group, which influence the lipid membrane packing [11,16,18]. Likewise, the polar head of the phospholipids is of great importance, as it determines the negative or positive surface charge of the vesicles, thus defining their kinetic stability [19]. Electrostatic repulsion between the particles depending on the charge level and ionic strength of the medium, and their balance with the van der Waals attractive forces determine the particle aggregation kinetics, which affects the protection and release of the encapsulated compounds [20].

The efficiency of encapsulation, the control of leaks during storage, the kinetic stability over time and the release dynamics of the encapsulated compound in liposomal systems mainly depend on the membrane packing characteristics, such as the thickness, fluidity, permeability and rotational mobility of phospholipids and the packing density. The molecular interactions between the encapsulated compound and the vesicle components are highly relevant when explaining the mechanisms involved in both the compound load and release [21].

Liposomal encapsulation has been used to incorporate essential oils, or their constituents, into hydrophilic systems, such as hydrocolloid aqueous solutions for film formation, in order to limit the emulsion destabilisation phenomena (droplet flocculation, coalescence or creaming); this leads to an accelerated loss of the active compound during the film drying step, with the subsequent reduction in the bioactivity of the material [22,23]. Previous studies into starch–gellan films containing liposome-encapsulated thyme essential oil [8] and chitosan films with liposome-encapsulated eugenol or cinnamon leaf essential oil [23] reported an increase in the retention of the active compound in the dried polymer films, with a notable increase in the bioactivity of the material. However, a partial release of the active compound during the film drying step and the formation of new phospholipid associations were also reported. The structural rearrangement of lipid membranes is associated with the changes in the molecular interactions in the system associated with water loss, which leads to a new thermodynamic state with minimal free energy (lyotropic mesomorphism) [15,16].

The characteristics of liposomes, depending on their composition, as well as their kinetic stability, have been widely studied in aqueous media [24,25,26,27]. However, the stability of liposomal structures and their encapsulation capacity when incorporated into polymer solutions and matrices are still issues to be addressed, given how interesting they are for the purposes of developing active polymeric materials. In this sense, the objective of this study was to characterise liposomal systems obtained with different lipid compositions (different lecithins), encapsulating carvacrol, by using different processing methods (rotor–stator homogenisation and sonication). Likewise, the different liposomal systems were incorporated into two types of PVA polymeric matrices (fully (F) and partially (P) hydrolysed) to obtain active films by casting. The degree of carvacrol retention in the films after the film drying step was analysed as well as their microstructure. The efficiency of the different lecithin at promoting carvacrol retention in the film was analysed in each polymeric matrix in order to identify the best lecithin–PVA combination for the obtaining of carvacrol-rich films.

Liposomes obtained with phosphatidylcholine-rich soy lecithin (SL-PC) were the most effective at maintaining the stability of carvacrol emulsion during film formation, which led to the maximum level of carvacrol retention in the films. Likewise, P-PVA systems were less sensitive to the emulsion destabilisation effects and to the kind of lecithin used in the liposome formation. These materials are of great potential for the production of active films for food packaging applications.

## 2. Results and Discussion

### 2.1. Liposomal Characteristics

Figure 1 shows the different size distributions of newly prepared liposomal systems. All treatments presented a main peak, between 179 and 294 nm, with a very low-intensity peak at around 4000 nm. Thus, nanosized particles were obtained in all cases, with a small ratio of microdroplets probably attributed to the aggregation of other lipid molecules present in the raw lecithin. The distribution curves of the carvacrol-free liposomes generally presented a small shoulder at the onset of the main curve, whereas the incorporation of CA promoted a narrower distribution, without the initial shoulder. This suggests structural changes in the liposomal membranes, promoted by the carvacrol–lipid interactions. These changes implied a decrease in both the hydrodynamic diameter (D_H_) and polydispersity index (PDI) (Table 1).

Table 1 shows the values of the hydrodynamic diameter (D_H_), the polydispersity index (PDI) and the ζ-potential of the different liposomal systems, with different lecithin and carvacrol ratios, obtained by rotor–stator homogenisation and sonication. The influence of the factors, type of lecithin, carvacrol-loaded ratio (CA: LEC) and preparation method on the different properties of liposomes (response variables) was statistically analysed, and the significance level for each one is also shown in Table 1.

The hydrodynamic diameter of liposomes ranged between 179 and 294 nm and the amplitude of the diameter distribution (PDI) between 0.17 and 0.36. The values of these parameters were significantly affected (95% confidence level) by all the factors considered. In general, sonication promoted the formation of smaller liposomes with a lower PDI, in agreement with that previously reported by other authors for this method, which produces homogeneous dispersions of nanometric liposomes [18,26,28]. The sonication effects are based on the cavitation of the bubbles, which provokes a localised, short-lived pressure increase, as well as microstreaming effects and shock waves that rupture lipid vesicles and promote the formation of more homogeneous systems [29]. A linear decrease in the particle size and polydispersity of liposomes as a result of increasing the time and intensity of sonication has previously been reported [18,26].

The type of lecithin also had a significant effect on the particle size and distribution. The liposomes from SL-PC were the smallest with the narrowest distributions (D_H_: 195–250 nm; PDI: 0.17–0.32), followed by those of SL (D_H_: 219–293 nm; PDI: 0.27–0.33) and SFL (D_H_: 179–294 nm; PDI: 0.25–0.36). The phospholipid composition of each lecithin gave rise to distinct liposomal topologies [30]. The high concentration of phosphatidylcholine (PC: 74%) of SL-PC could lead to liposomes with more homogeneous, cohesive and stable lipid membranes. PC has relatively large head groups, which promotes the spontaneous formation of lipid bilayers, defining the specific membrane curvature and the shape of the liposomes [31]. In contrast, the SL and the SFL, with a lower concentration of PC, present other phospholipid and nonphospholipid impurities (amino acids, free fatty acids), giving rise to liposomal membranes of mixed composition, probably less cohesive and more fluid, which is generally reflected in a higher D_H_ and a greater variation in the size distribution (higher PDI). As is well known, the physicochemical properties of liposomes are dictated by their lipid composition, since each phospholipid has a different molecular structure and geometry that mark the structural characteristics of the liposomes. The specific membrane packing parameter is mainly determined by the size of the head groups and hydrophobic tails [31]. Thus, longer-chain phospholipids generate liposomes with good cohesion due to the greater attraction force of van der Waals between the chains.

Regardless of the CA:LEC ratio in the systems, the incorporation of carvacrol led to a reduction in the liposomal size, which was especially notable in the SL and SFL liposomes, whereas the size of SL-PC liposomes was less sensitive to the carvacrol load. The size reduction provoked by the carvacrol load is coherent with changes in molecular interactions within the membrane introduced by the phenolic compound. These promoted membrane reorganisations and subsequent changes in D_H_ and PDI. Similar results were reported for liposomes from sunflower seed lecithin, with 20% of PC-encapsulating carvacrol [23]. In contrast, the liposome encapsulation of eugenol led to an increase in liposomal size due to different changes in molecular interactions associated with its molecular structure [11,12]. The changes in both size and ζ-potential depended on the molecular structure (size and shape), polarity and chemical affinity of the loaded phenolic compounds [32,33]. Thus, phenolic compounds with several phenolic hydroxyl groups can interact cooperatively with lipid polar heads, inducing conformational changes in the membranes [34]. In contrast, more hydrophobic molecules can interact with the acyl chains of the bilayer inside the lipid membrane, which could provoke a decrease in the liposome size. Furthermore, changes in the fluidity of the membrane can explain the liposome size change [35,36].

The values of the ζ-potential were highly negative in the obtained liposomal dispersions, which is associated with a high degree of stability, since repulsive electrostatic forces are promoted against attractive van der Waal forces [19]. The ζ-potential was significantly affected by the type of lecithin and CA load (*p* < 0.05). In this sense, different values were obtained for each type of lecithin; SL liposomes (−52, −57 mV) had the highest absolute value, followed by SL-PC liposomes (−43, −50 mV) and SFL liposomes (−33, −38 mV). CA incorporation enhanced the negative charge of the liposomes, probably due to the membrane changes associated with the carvacrol load, which can promote the greater exposure of negative charges.

Differences in the ζ-potential values of each lecithin liposome must be attributed to its specific composition, since it is mainly associated with the negative charge of the PO_4_^−3^ head group of the phospholipids, and the presence of impurities, such as free fatty acids, which are associated in the lipid membrane [18]. In this sense, the presence of anionic lipids, such as phosphatidic acid (PA) and phosphatidylinositol (PI), can induce an increase in the electronegativity of liposomes [37], which agrees with the highest ζ-potential of SL that contains the highest ratio of PA and PI (Table 3).

### 2.2. Liposomal Stability

The stability of liposomal systems was evaluated through the changes in their size and charge throughout time. Figure 2 shows the values of the hydrodynamic diameter and the ζ-potential of the different liposomes after one, four and eight days of storage at 4 °C in darkness. Every liposomal system exhibited very slight changes in both D_H_ and ζ-potential, regardless of the presence or absence of carvacrol. Therefore, no destabilisation of the colloidal systems during the evaluated period could be deduced. The high surface charge (ζ-potential ≥ |−30 mV|) of liposomes favours the stability of the colloidal system due to the electrostatic repulsion of individual particles [38,39].

The slight changes in the D_H_ and in the ζ-potential over time (Figure 2) indicated the liposomal dynamic character, promoted by the small fluctuations in the particle environment and the dynamic equilibrium between associated molecules in the colloidal system. This dynamic equilibrium is responsible for the typical characteristics of the liposomal system, such as the fluidity and flexibility of the lipid bilayer, and the continuous membrane reorganisation in response to the mobility and molecular rotation rate of lipids and phospholipids [25,40]. Although no significant differences between the stability of CA-free and CA-loaded liposomes were detected, other authors [41,42] reported that the spatial rearrangement of the liposomal membrane, caused by carvacrol, increased the system stability due to the reduction of the repulsive forces among the head groups of phospholipids; this decreased the degree of mobility of the hydrophobic tails and led to closer molecular packing.

Stability studies of soybean and salmon lecithin liposomal systems, encapsulating polyphenols and curcumin, respectively, indicated that the significant changes in liposomal characteristics (which occurred after around five weeks) could be associated with the oxidation and hydrolytic degradation of the unsaturated fatty acids of phospholipids, causing a decrease in the absolute value of ζ-potential and an increase in the D_H_ of the particles [25,43].

### 2.3. Carvacrol Content and Film Microstructure of PVA Matrices with Liposomes

The different CA-loaded liposomes were incorporated into aqueous solutions of fully (F-PVA) and partially (P-PVA) hydrolysed PVA to obtain carvacrol-loaded films by casting. The different efficiency of each liposomal system and polymer matrix to retain the volatile carvacrol was evaluated through the analysis of the CA content in the dry films (Table 2) and the film microstructure (Figure 3). Very big losses of volatile compounds, such as carvacrol, have been observed during the film-forming step (drying) from the casting of aqueous polymer solutions with emulsified organic compounds. Throughout the drying step, emulsion destabilisation mechanisms (flocculation, coalescence and creaming) take place in line with water evaporation. As a consequence, large droplets of the organic compounds migrate to the film surface and evaporate, together with water molecules, by the steam drag effect [44], which constitutes the main reason for the losses of these compounds in the cast films.

The liposome encapsulation of volatile compounds, such as carvacrol, can reduce these losses due to the stabilising effect of lipid membranes. Nevertheless, lipid associations, such as liposomes, are also very sensitive to water availability, exhibiting lyotropic mesomorphism and phase transitions as a function of the water content [16]. This implies that when liposomes are incorporated into aqueous film-forming solutions, e.g., to obtain films loaded with carvacrol, they will also suffer changes in their association. Likewise, the molecular interactions of lecithin lipids and carvacrol with the film polymer would play an important role in the properties and in the stability of the aqueous emulsion and, therefore, in the final content of the volatile compound in the film. In fact, in the presence of a polymer, different structural changes in liposomes could also be expected during the drying of the film, as well as the release of carvacrol from the liposomes into the polymer-rich phase. The changes will progress in line with the process of water evaporation during film formation. These dynamic changes will give rise to a final carvacrol content in the films, resulting from both the different destabilisation phenomena that have occurred and the final molecular rearrangement according to the chemical affinity and interactions.

Table 2 shows the amount of carvacrol retained in the film with respect to the initially incorporated amount (% retention) for the two types of PVA polymers and the different liposomal systems (with different lecithin, carvacrol loads and preparation methods). The CA retention was significantly (*p* < 0.05) affected by the type of lecithin and by the liposome preparation method, as well as by the type of PVA. Films with SL-PC liposomes had the highest final CA content, probably due to the lipid composition of lecithin, rich in phosphatidylcholine. In fact, a more homogeneous, more cohesive and less fluid liposomal membrane could be obtained for SL-PC [45], with greater control of the CA leakage. Furthermore, the greater packing density of lipid bilayer could be less susceptible to both the phenomenon of destabilisation and to the structural modifications associated with water loss during film drying. In general, the liposomes obtained by sonication also showed slightly greater CA retention in the films, probably due to their smaller size and lower PDI in comparison with those obtained by rotor–stator homogenisation. This could favour their kinetic stability, reducing the rate of particle aggregation and coalescence [46].

On the other hand, partially hydrolysed PVA presented the highest CA retention values (between 54% and 74%), compared to fully hydrolysed PVA, with percentage retention values between 22% and 57%. It is worth highlighting that during the film formation process, liposomal systems present carvacrol leakage, due to the lyotropic mesomorphism of liposomes [8,47]. Thus, the degree of carvacrol retention would depend not only on liposomal encapsulation systems, but also on the interaction of the active compound released with the polymer matrix. P-PVA has residual acetyl groups (12%) in the polymer chain, which acquire a state of resonance and provide the matrix with a negative charge [48]. This donor electron pair provides the P-PVA chains with the Lewis base character, which promotes the formation of Lewis adducts with carvacrol (Lewis acid). This mechanism enhances the chemical affinity between carvacrol and the polymer chains, increasing their retention in the film [5,48,49]. In this sense, it was possible to observe that the type of lecithin had less influence on the CA retention for P-PVA matrices, since the carvacrol released from liposomes could be better entrapped through its bonding to the P-PVA chains. In contrast, the type of lecithin greatly affected the carvacrol retention in the F-PVA matrix with a lower chemical affinity for carvacrol. The carvacrol retention in this polymer matrix markedly reflects the different stabilising capacity of each lecithin, SL-PC being the most effective and SFL the least at stabilising the carvacrol-loaded liposomes during film formation.

Figure 3 shows FESEM micrographs of the cross-section of PVA films (F and P) with liposomes from different kinds of lecithin, loaded or not with carvacrol. These micrographs reveal qualitative differences related to the arrangement of the components in the samples. A first remarkable feature was the smooth appearance of P-PVA films with free carvacrol (nonliposome encapsulated) (Figure 3a), in contrast to the carvacrol droplets observed in the F-PVA matrix (Figure 3b). This fact demonstrates the good integration of this phenolic compound in P-PVA matrix, without visible dispersed droplets, while the F-PVA films presented a clear phase separation. This was attributed to the formation of Lewis adducts between the phenolic compound and the negatively charged chains of P-PVA, as previously described [28,29]. Nevertheless, the incorporation of carvacrol-free liposomes into P-PVA and F-PVA matrices led to heterogeneous matrices that exhibited a dispersed phase of altered liposomes (Figure 3c,d, respectively).

Different microstructural features were also observed for films with carvacrol-loaded liposomes as a function of the kind of lecithin in agreement with the different stabilising effects of the lipid membranes during film formation. The more stable membranes exhibit a lower level of disruption during the water evaporation step, remaining less altered in the films and entrapping carvacrol more efficiently. In the P-PVA matrix with carvacrol- loaded SL-PC liposomes (Figure 3e), a relatively fine dispersion of lipid particles (about 1 µm or smaller), partially altered in shape, could be observed as a result of mesomorphic transitions in the lipid membranes. According to the high chemical affinity between carvacrol and P-PVA chains, the leakage of carvacrol from the liposomes would not imply the presence of free carvacrol droplets. The SL-CA and SFL-CA liposomes also appeared dispersed in the P-PVA matrix (Figure 3g,i), similar in appearance to those of the SL-PC liposomes, but exhibiting larger particle traces and more coalescing droplets.

In contrast, much larger particles could be observed in F-PVA films with carvacrol- loaded liposomes (Figure 3f,h,j), indicating the greater progress of coalescence during film drying. For SFL liposomes, a particularly high level of lipid separation was observed in the matrix, thus showing the poor stabilising capacity of this type of lecithin. The microstructural observations agree with the different carvacrol retention in the films associated with the stabilising capacity of the lipid membranes (Table 2). This ability was affected by polymer–lipid interactions, linked to the degree of hydrolysis of the PVA chains and the lipid composition of the liposome membrane. Despite the slightly higher surface charge of SL liposomes, the high ratio of PC in the SL-PC membrane seems to produce more stable structures in aqueous PVA solutions that provide emulsified systems with greater stability during the water evaporation step, helping to keep the carvacrol contained in the dry films. SFL liposomes, with a more heterogenous lipid composition and lower ζ-potential values, were less effective at stabilising emulsified carvacrol in aqueous PVA solutions.

## 3. Materials and Methods

### 3.1. Materials

The liposomal vesicles were obtained using different types of lecithin (LEC); soybean lecithin enriched (74%) in phosphatidylcholine PC (SL-PC) (Lipoid S75, Lipoid GmbH, Ludwigshafen, Germany), soy lecithin (SL) (Guinama, Prague, Czech Republic) and sunflower lecithin (SFL) (Cargil S.L.U., Barcelona, Spain), whose phospholipid compositions are shown in Table 3. Carvacrol was purchased from Sigma-Aldrich (Steinheim, Germany). Two types of poly (vinyl alcohol) (PVA): fully hydrolysed (F: Mw 89,000–98,000; 99–99.8% hydrolysed) and partially hydrolysed (P: Mw 13,000–23,000; 87–89% hydrolysed) (Sigma-Aldrich, Steinheim, Germany), were used as polymeric matrices to develop the films. Phosphorus pentoxide salt (P_2_O_5_) and UV-grade methanol were supplied by Panreac Química S.A. (Barcelona, Spain).

### 3.2. Liposome Production

Lecithin (LEC) (5% *w*/*w*) was initially dispersed in distilled water using magnetic stirring for 30 min at 800 rpm. The CA was incorporated into the mixture using CA:LEC ratios of 0:1, 0.5:1 and 1:1. To obtain nanoliposomes in the initial dispersion, two methods were applied: homogenisation using a rotor–stator mixer (IKA digital T25 UltraTurrax, Staufen, Germany), at 1100 rpm for 5 min and sonication (35 kHz) for 10 min with 1 s pulses, using an ultrasonic device (Vibra Cell, Sonics & Material, Inc. USA), maintaining the sample in an ice bath to avoid heating. All samples were kept in the dark at 4 °C till their use.

### 3.3. Liposome Properties

The particle size distributions, the hydrodynamic diameter (D_H_) and the polydispersity index (PDI) were obtained in triplicate using the dynamic light scattering-DLS technique by means of a Zetasizer NanoZS, using DTS 1070 cell (Malvern Instruments Zen 3600, United Kingdom). The ζ-potential was calculated by means of Henry’s equation from the electrophoretic mobility of the vesicles. Measurements were taken at 25 °C, after 100-fold dilution of the liposomal suspension in distilled water. The measurements were taken in triplicate at different times (after 1, 4 and 8 days) in order to evaluate the physical stability of the sample throughout time.

### 3.4. Preparation of Films

The PVA films were obtained by casting. Polymer solutions (F-PVA 5 wt.% and P-PVA 10 wt.%) were prepared in distilled water using magnetic stirring (1200 rpm) at 100 °C for 3 h. Liposomes were added to the polymer solutions to achieve a lecithin–polymer ratio of 10%. Later, different concentrations of CA were added (0, 5 or 10 wt.% with respect to the polymer) [47]. All formulations were degassed by using a vacuum pump before being evenly spread onto Teflon plates of 150 mm in diameter, using a constant equivalent mass of polymer per plate of 2 g. The films were dried under controlled temperature (25 ± 2 °C) and relative humidity (54 ± 2%) for 48 h. The analyses of final CA content in the films and microstructural analyses were carried out in films conditioned at 0% RH by using P_2_O_5_.

### 3.5. Final Carvacrol Content in the Films

The analysis of CA retained in the different film formulations was carried out by spectrophotometry after the extraction of CA from the dry films. Thus, film samples (4 cm^2^) were immersed in 50 mL of a 50% aqueous solution of UV-grade methanol and kept under stirring (300 rpm) for 48 h at 23 °C. The absorbance (A) of the aliquots was measured at 274 nm by using a spectrophotometer (Evolution 201 UV-Vis, Thermo-Fisher Scientific Inc., Waltham, MA, USA). The CA-concentration (C) in the films was determined by means of a standard curve, which was obtained from solutions with different concentrations of carvacrol (10–50 µg/mL) in the same solvent (C = 63.61A, R^2^ = 0.998). As backgrounds, the corresponding extracts obtained under the same conditions from the CA-free films were used. The final content of CA (%) in the films was calculated as the ratio between the amount of CA extracted from the film with respect to the corresponding amount of initially incorporated CA.

### 3.6. Microstructure of Films

The cross-section microstructure of the films was evaluated by using a field emission scanning electron microscope (FESEM) (ZEISS^®^, model ULTRA 55, Jena, Germany) and an acceleration voltage of 2 kV. To obtain the cross-section micrographs of the films, the samples were immersed in liquid nitrogen, cryofractured and coated with platinum before obtaining the images.

### 3.7. Statistical Analysis

The statistical analysis of the data was carried out using Statgraphics Centurion XVI.II. The results were submitted to a multifactor ANOVA to estimate the significance of the factors that influence the response variables. Fisher’s least significant difference (LSD) was used with a confidence level of 95%.

## 4. Conclusions

The composition of lecithin of differing origins affected the stability of the carvacrol-loaded liposomes incorporated into the PVA films obtained by casting the aqueous polymer solutions. The lyotropic mesomorphism of lipid associations and the carvacrol leakage occurred to different degrees during the film drying step, depending on the lipid composition of the membrane and its resulting surface charge. Lipid interactions with the polymer also play an important role in liposomal stability and the final carvacrol retention in the films. Liposomes obtained with phosphatidylcholine-rich soy lecithin (SL-PC) were the most effective at maintaining the stability of the carvacrol emulsion during film formation, which led to the highest carvacrol retention, whereas SFL gave rise to less stable systems and greater carvacrol losses. Likewise, systems with partially hydrolysed PVA were less sensitive to the emulsion destabilisation effects and to the kind of lecithin used in the liposome membranes due to the greater chemical affinity and bonding capacity between carvacrol and polymer chains. Therefore, P-PVA with carvacrol-loaded SL-PC liposomes with a carvacrol retention capacity of about 70% in films has great potential to produce active films for food packaging applications. In this sense, the antioxidant and antimicrobial properties of the potentially active materials with carvacrol should be evaluated in further studies, both through in vitro tests and using different food matrices.

## Figures and Tables

**Figure 1 molecules-26-01589-f001:**
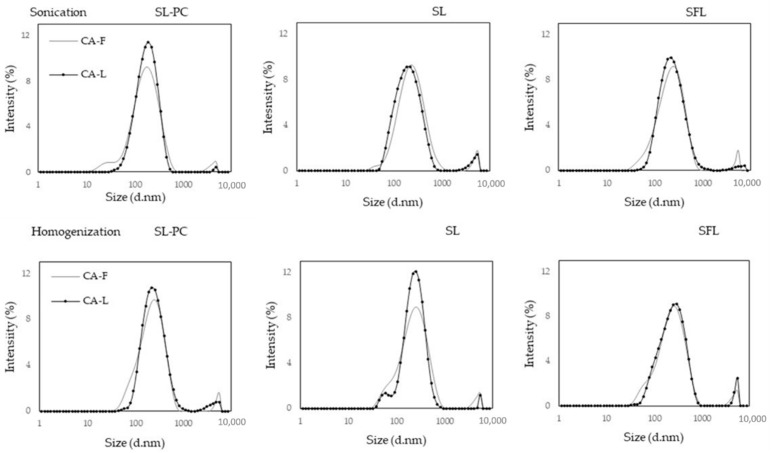
Particle size distribution of carvacrol-free (CA-F) and carvacrol-loaded liposomes (CA-L) (carvacrol–lecithin ratio CA:LEC: 1:1) using different lecithins and preparation methods.

**Figure 2 molecules-26-01589-f002:**
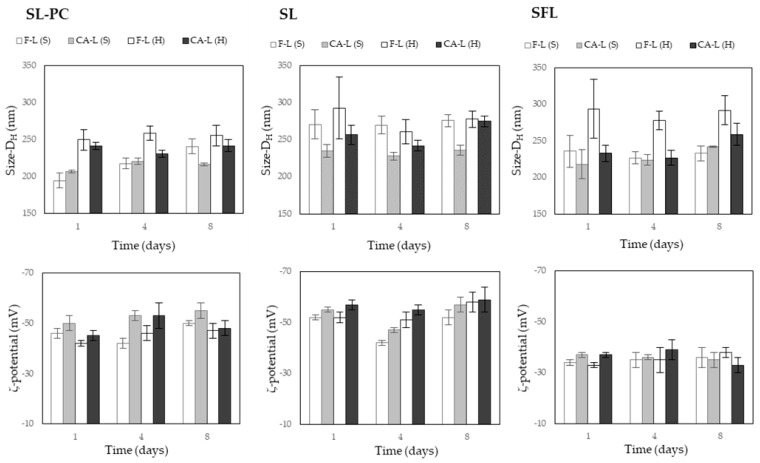
Development over time of the hydrodynamic diameter (D_H_) and the ζ-potential of liposomes from soy lecithin enriched in phosphatidylcholine (SL-PC), soy lecithin (SL) and sunflower lecithin (SFL), carvacrol-free (F-L) or CA-loaded (CA-L) (CA: LEC: 1:1) liposomes, obtained by sonication (S) or rotor-stator homogenisation (H).

**Figure 3 molecules-26-01589-f003:**
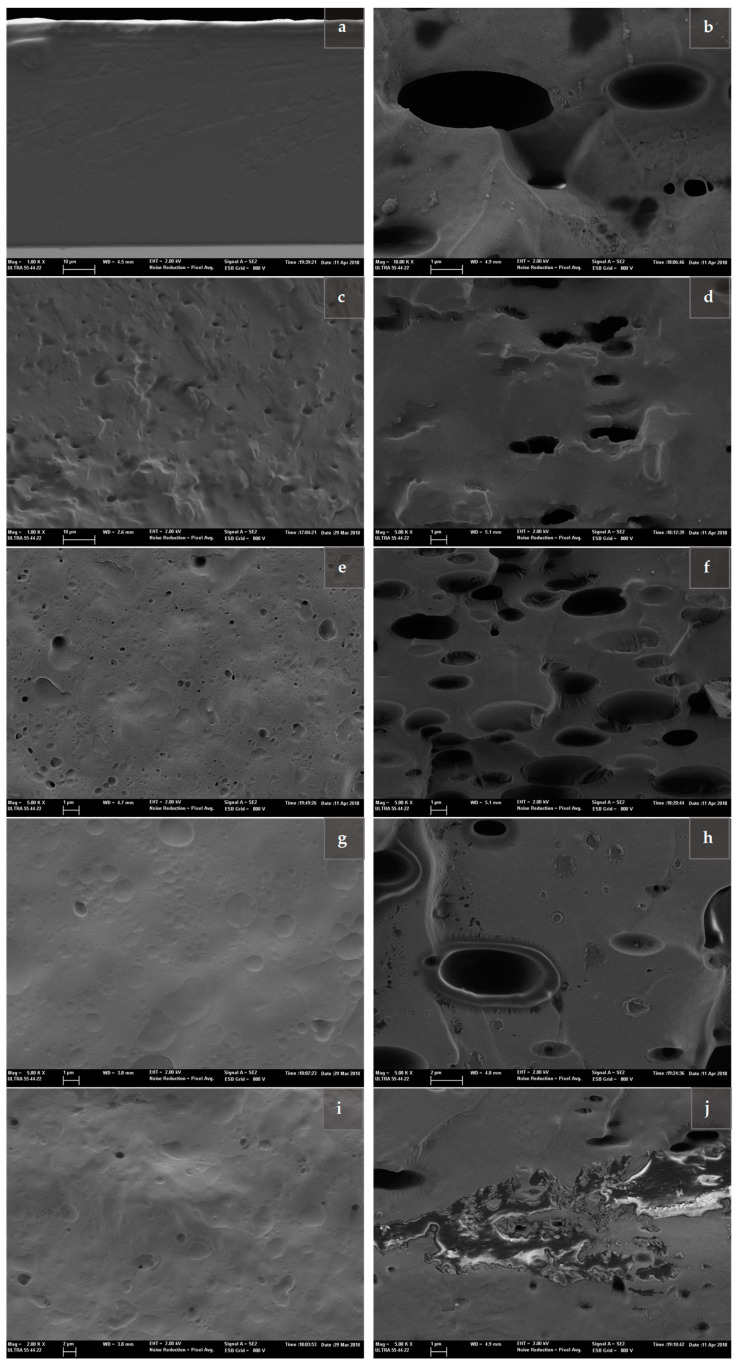
FESEM micrographs of the cross-section of F-PVA and P-PVA films with CA-loaded liposomes (CA: LEC: 1:1) using different types of lecithin (soy lecithin enriched in phosphatidylcholine PC (SL-PC), soy lecithin (SL) and sunflower lecithin (SFL)) and the sonication method. (**a**) P-PVA with free CA; (**b**) F-PVA with free CA; (**c**) P-PVA with CA-free SL-PC; (**d**) F-PVA with CA-free SL-PC; (**e**) P-PVA with CA-loaded SL-PC; (**f**) F-PVA with CA-loaded SL-PC; (**g**) P-PVA with CA-loaded SL; (**h**) F-PVA with CA-loaded SL; (**i**) P-PVA with CA-loaded SFL; (**j**) F-PVA with CA-loaded SLF.

**Table 1 molecules-26-01589-t001:** Properties (hydrodynamic diameter (D_H_), polydispersity index (PDI) and the ζ-potential (ζ) of liposomal systems obtained with the different types of lecithin (soy lecithin enriched in phosphatidylcholine (SL-PC), soy lecithin (SL) and sunflower lecithin (SFL)), using different carvacrol–lecithin ratios (CA:LEC: 0:1, 0.5:1, 1:1) and preparation methods (sonication and rotor–stator homogenisation).

LEC	CA:LEC	Method					
		Sonication			Homogenisation		
		D_H_ (nm) *	PDI	ζ (mV)	D_H_ (nm) *	PDI	ζ (mV)
SL-PC	0:1	195 (10)	0.28 (0.01)	−46 (2)	250 (14)	0.29 (0.03)	−43 (1)
	0.5:1	211 (1)	0.24 (0.02)	−47 (2)	249 (12)	0.32 (0.01)	−44 (1)
	1:1	207 (2)	0.17 (0.01)	−50 (3)	242 (5)	0.19 (0.01)	−45 (2)
SL	0:1	271 (20)	0.31 (0.03)	−52(1)	293 (42)	0.32 (0.04)	−52 (2)
	0.5:1	219 (8)	0.32 (0.01)	−53 (1)	232 (30)	0.33 (0.01)	−56 (2)
	1:1	235 (9)	0.27 (0.05)	−55 (1)	257 (13)	0.31 (0.01)	−57 (2)
SFL	0:1	236 (22)	0.30(0.02)	−34 (1)	294 (40)	0.35 (0.03)	−33 (1)
	0.5:1	192 (5)	0.35 (0.02)	−38 (1)	179 (7)	0.36 (0.03)	−37 (1)
	1:1	218 (20)	0.25 (0.03)	−37 (1)	233 (11)	0.36 (0.01)	−37 (1)

* Value corresponds to the main peak of distribution size (containing ≥94% of particles). Factors with a statistically significant effect with a 95% confidence level. DH: method (*p* = 0.0003), lecithin (*p* = 0.0048), CA:LEC (*p* = 0.0001); PDI: method (*p* = 0.0001), lecithin (*p* = 0.0002), CA:LEC (*p* = 0.0001); ζ: lecithin (*p* = 0.0001), CA:LEC (*p* = 0.0001).

**Table 2 molecules-26-01589-t002:** Carvacrol retention (ratio of determined carvacrol with respect to the incorporated amount, in percentage) in dry PVA films for fully (F) and partially (P) hydrolysed polymer and different carvacrol-loaded liposomal systems.

LEC	CA: LEC	CA Retention (%)			
		F-PVA		P-PVA	
		Sonication	Homogenisation	Sonication	Homogenisation
SL-PC	0.5:1	54 (2)	53 (4)	74 (2)	71 (5)
	1:01	57 (3)	54 (3)	67 (3)	65 (3)
SL	0.5:1	47 (1)	41 (1)	66 (1)	64 (1)
	1:01	45 (4)	46 (4)	66 (7)	62 (5)
SFL	0.5:1	28 (1)	22 (2)	61 (7)	54 (8)
	1:01	27 (2)	24 (4)	63 (3)	60 (3)

Factors with a statistically significant effect with a 95% confidence level. CA retention: method (*p* = 0.0036), lecithin (*p* = 0.0001), PVA type (*p* = 0.0001) and the interaction between lecithin and PVA (*p* = 0.0001).

**Table 3 molecules-26-01589-t003:** Phospholipid composition of soybean lecithin enriched in phosphatidylcholine (SL-PC), soy lecithin (SL) and sunflower lecithin (SFL).

Phospholipid	% (*w*/*w*)		
	SL-PC	SL	SFL
PC-phosphatidylcholine	74	14	12
PI-Phosphatidylinositol		12	4
PE-Phosphatidylethanolamine	11	10	11
LPC-Lysophosphatidylcholine	3	0	3
PA-Phosphatidic acid	1	4	2

## Data Availability

Data is contained within the article.

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
