# Peer review of "Liposomal Encapsulation of Carvacrol to Obtain Active Poly (Vinyl Alcohol) Films"

_molecules, 2021, doi:10.3390/molecules26061589_

Round 1

Reviewer 1 Report

Please take into consideration the recommendations from the attached file.

Reviewer 2 Report

Dear Editor, dear Authors, Johana Andrade et al. submitted a paper on the use of lecithin from different origin and composition (soy lecithin (SL), soy lecithin enriched with phosphatidylcholine (SL-PC) and sunflower lecithin (SFL)) for liposomal encapsulation of carvacrol (CA) at different CA:LEC ratio ((CA:LEC 0:1, 0.5:1, 1:1) by sonication or rotor- stator homogenization methods. The authors aim is to introduce the vesicles into two types of PVA polymeric matrices (fully (F) and partially (P) hydrolyzed) to obtain active films by casting for food packaging. Therefore, the authors have characterized their resulting liposomes by Dynamic Light Scattering (DLS) and by performing the ζ-potential measurements as well as their stability after 1, 4 and 8 days of storage at 4 °C in dark condition. Moreover, they have also evaluated the degree of carvacrol retention by the analysis of the CA content in the dry films, as well as their microstructure. The authors’ results demonstrated that the CA retention was significantly affected by the origin of lecithin used (SL-PC show the best results) and by preparation method of liposomes, as well as by the type of PVA (P-PVA here show better results). For my opinion, the authors wrote an interesting paper, the manuscript is well written, and results are well supported with experimental evidence. Therefore, I believe that the manuscript can be accepted for publication in Molecules Journal. However, I suggest that the authors include in the manuscript a scheme (and a graphical abstract) summarizing the different steps used for the preparation of liposomes and films. I have also minor comments to the authors that should be revised.

  • Page 1, line 35 “matrices has been extensively studied as an strategy as an strategy for developing active food packaging”. Please correct a strategy not an.
  • Page 2, line 87 “Likewise the different liposomal systems were…” add a comma after Likewise.
  • Page 5, line 206 “The slight changes in the DH and in the ζ-potential over time (Figure 3)” The authors are discussing here Figure 2 and not 3, please revise.
  • I suggest that the authors include in the manuscript a scheme showing the different steps used for the preparation of liposomes and films.

Sincerely Yours,
